# Design of an Intermittent Herbicide Spray System for Open-Field Cabbage and Plant Protection Effect Experiments

**Shenyu Zheng** [1,2,†], **Xueguan Zhao** [3,†], **Xinwei Zhang** [2], **Hao Fu** [1], **Kechuan Yi** [2,*] **and Changyuan Zhai** [1,3,*]

1    Intelligent Equipment Research Center, Beijing Academy of Agriculture and Forestry Sciences, Beijing 100097, China
2    College of Mechanical Engineering, Anhui Science and Technology University, Fengyang 233100, China
3    National Engineering Research Center for Information Technology in Agriculture, Beijing 100097, China
*    Correspondence: yikechuan@sina.com (K.Y.); zhaicy@nercita.org.cn (C.Z.);
     Tel.: +86-55-0673-2023 (K.Y.); +86-10-5150-3886 (C.Z.)
†    These authors contributed equally to this work.

**Abstract:** To address the problem of herbicide residues exceeding the safety standard due to continuous spraying of herbicides on open-field cabbage, we propose an intermittent weed spraying control method integrating cabbage position, cabbage canopy size, and spraying machine operation speed. It is based on an early-stage cabbage target identification method obtained in the early stage and the operation requirements in open-field cabbage. Built with a C37 controller, a stable pressure spray system and an intermittent weed spraying control system for open-field cabbage, an integrated system was designed. Experimental verification was carried out through measurement indexes such as spraying precision, herbicide saving rate, herbicide efficacy, and herbicide residue. Since the industry is faced with a status quo of a lack of relevant operational norms and national standards for the precise weed spraying operation mode, this paper provides a relatively perfect experiment and evaluation method for this mode. The experimental results on the accuracy of weed spraying at different speeds showed that the mean absolute error (MAE), root mean square error (RMSE), and average spray cabbage coverage rate (ASCCR) of intermittent weed spraying increased, but the average effective spray coverage rate (AESCR) decreased with increasing operation speed. When the working speed was 0.51 m/s, the MAE and RMSE of intermittent weed spraying were less than 2.87 cm and 3.40 cm, respectively, and the AESCR was 98.4%, which verified the feasibility of operating the intermittent weed spraying of cabbage. The results of a field experiment showed that the average weed-killing rate of intermittent weed spraying for open-field cabbage was 94.8%, and the herbicide-saving rate could reach 28.3% for a similar weeding effect to that of constant-rate application, which not only met the needs of intermittent weed spraying in open-field cabbage but also had great significance for improving the herbicide utilization rate. Compared with the constant-rate application method, the herbicide residue concentration detected using intermittent weed spraying for cabbage decreased by 66.6% on average, which has important research significance and application value for ensuring the normal growth of crops and the safety of agricultural products.

**Keywords:** plant protection machinery; sensor; spray rod type; herbicides; pesticide residues; weed killing rate

## 1. Introduction

Weeds compete with crops for sunshine, water, space, and nutrients, which results in a decline in crop yield and quality [1] and causes very large losses to the agricultural economy. Currently, there are two main weeding methods, namely, mechanical weeding and chemical weeding. Mechanical weeding has poor adaptability to topography and agronomy [2,3], high energy consumption [4], and low operation efficiency [5], and some weed types cannot be completely eradicated by mechanical weeding [6]. Chemical weeding is the most widely

used weeding method by farmers worldwide [7–9]. However, existing chemical weeding methods utilize constant-rate application, which easily leads to the problem of excessive herbicide residues in crops, affects the normal growth of crops and poses a great threat to the safety of agricultural products. How to apply spraying technology to successfully kill weeds and reduce the adverse effects of herbicides on crops and the environment is still an urgent problem to be solved [10].

Researchers have carried out related studies on prescription application [11–13], variable application [14–16], atomization droplet characterization [17,18], spraying uniformity, and operation stability [19]. For crops with large row and plant spacing, such as cabbage, corn, and cotton, when herbicides are continuously sprayed in the early growth stage, the liquid sprayed on the plants will cause waste and bring pesticide residue problems to the crops, which will affect the normal growth of crops. With the development of modern green agriculture, technology for weed spraying in a target-oriented manner has been proposed, which combines target identification with spraying operation and controls the nozzle to aim at the target position according to the operation speed of the spraying machine, which can greatly reduce the spraying probability of crop position in the field and achieve the purpose of reducing the amount of herbicide applied. At the same time, it is of great significance to improve the utilization rate of herbicides. Compared with a constant-rate application method, the herbicide efficacy of a target-oriented weed spraying method is similar, and the herbicide residues on crops are greatly reduced, which has important research significance and application value for ensuring the safety of agricultural products.

In terms of target identification, researchers have carried out accurate weed spraying operations from the perspective of weed identification [20–22]. For example, Sharpe et al. [23] used a YOLOv3-tiny network to detect only goosegrass in strawberry and tomato fields to develop accurate herbicide application equipment. Liu et al. [24] used convolution neural networks (CNNs) to classify and spray only two weeds, spotted spurge and Shepherd's purse, in strawberry fields. Farooque et al. [25] used CNNs to detect and spray only Chenopodium album L. and Spergula arvensis L. in potato fields. These studies can meet certain weeding requirements by identifying weeds for accurate weed spraying. However, due to the small number of weeds in the dataset, they cannot meet the complex working environment in a field, which ultimately affects the quality of weed spraying operations. There are many kinds of weeds in a field, and the color discrimination between them and crops is low. It is difficult to accurately identify all kinds of weeds [26,27], and it is more difficult to identify weeds at the seedling stage, which requires extremely complex spray system hardware [28]. Accurate weeding and spray operations based on weed identification will greatly reduce the quality of weeding operations. Therefore, it is necessary to propose a new method to improve the weed-killing rate in complex field environments, ensure the normal growth of crops and reduce the deposition of liquid herbicide in crop positions.

At present, there is a lack of relevant technical specifications and national standards for the evaluation of the plant protection effect of precise weed spraying operations. Relevant studies on weed-killing rate statistics and herbicide residue monitoring when carrying out precise weed spraying operations are lacking. The important purpose of precise weed spraying is to achieve effective weed control. Based on meeting the requirements of herbicide application accuracy, it is of great research significance to analyze the actual efficacy under precise weeding operation further and track the dosage and residue of precise weed spraying.

This study aims to explore a weed spraying method based on cabbage identification and improve and optimize the evaluation of plant protection effects. Specifically, an intermittent weed spraying control model and intermittent weed spraying control system were established by integrating a weed spraying control method including cabbage position, cabbage canopy size, and spraying machine operation speed. In addition, the plant protection effect of intermittent weed spraying operation was evaluated on four aspects: spraying accuracy, herbicide saving rate, herbicide efficacy, and herbicide residue, which provides a reference for accurate weed spraying operation.

## 2. Design of an Intermittent Weed Spraying System for Open-Field Cabbage

### 2.1. Hardware Architecture Design

The overall structure of the open-field cabbage intermittent weed spraying system is shown in Figure 1, which includes an identification unit, a herbicide supply unit, and a spraying control unit. The identification unit includes a network camera and a Jetson Xavier NX; the network camera is used for collecting cabbage images, and the Jetson Xavier NX is used for processing the cabbage image information collected by the network camera and sending the cabbage information to a C37 controller through serial communication. The herbicide supply unit includes a herbicide tank, filter, engine, plunger pump, and nozzle, which provides stable herbicide supply pressure for the herbicide spray system under the drive of the engine power output shaft. The spraying control unit includes a C37 controller, encoder, pressure sensor, flow sensor, electric ball valve, and solenoid valve. Among them, the C37 controller is the lower machine of the spraying control unit, which can collect the data of the encoder, pressure sensor, and flow sensor in real time. According to the pipeline pressure obtained from the pressure sensor, the valve opening of the electric ball valve is adjusted to keep the pressure in the system pipeline stable. The longitudinal canopy size of the cabbage identified by the upper machine is received through a USB-CAN converter. At the same time, the controller converts the pulse signal of the encoder into the operation speed and moving distance, adjusts the delay time of the system to compensate for the lag of the system, and accurately controls the solenoid valve to be opened only between cabbage plants and rows. This achieves the intermittent weed spraying operation of cabbage in an open field and reduces herbicide residue on crops. The main hardware models and parameters of the system are shown in Table 1.

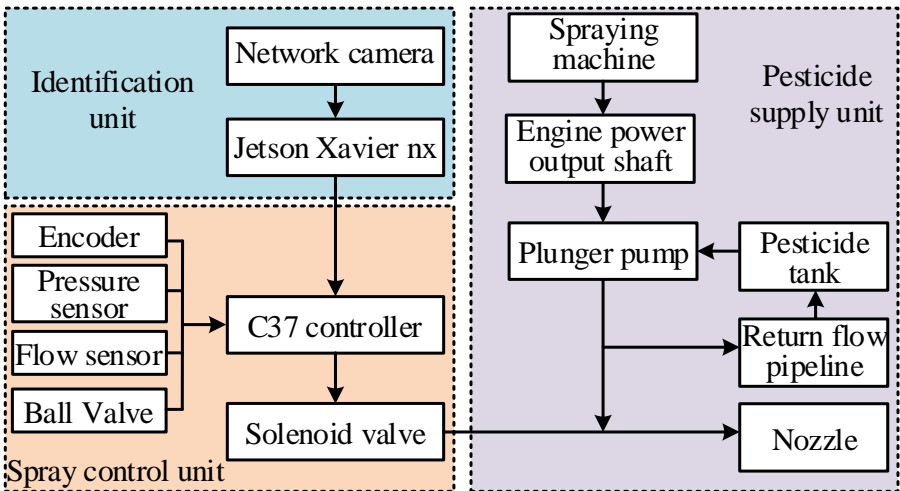

**Figure 1.** The overall structure of the field cabbage weed-spraying system.

**Table 1.** Main hardware models and parameters.

| Category | Model | Company | Main Parameters |
|---|---|---|---|
| Network camera | C930 | Shanghai Liuxiang Trading Co., Ltd., Shanghai, China | Frame rate 30 fps |
| Jetson Xavier NX | Developer Suite | Hunan Chuanglebo Intelligent Technology Co., Ltd., Changsha, China | Computational power 21 tflops |
| Encoder | HTS-5208 | Wuxi Hengte Technology Co., Ltd., Wuxi, China | Resolution 1000 |
| Pressure sensor | 131-B | Beijing Aosheng Automation Technology Co., Ltd., Beijing, China | Pressure range 0~2.5 MPa |
| Flow sensor | HI2144 | Yueqing Ponai Sensor Technology Co., Ltd., Wenzhou, China | Flow range 1~30 L/min |
| Electric ball valve | FRY-02T | Jiangsu Valve Ruiyi Valve Equipment Co., Ltd., Yancheng, China | Control voltage signal DC 0~10 V |
| Controller | C37 | Suzhou Hesheng Microelectronics Technology Co., Ltd., Suzhou, China | 12 PWM outputs, 4 pulse inputs and 6 analog inputs |
| USB-CAN converter | Isolated | Beijing Ledian Xinnan Technology Co., Ltd., Beijing, China | Support package mode |
| Solenoid valve | 2W-050-08ES | Yuyao NO.4 Instrument FACTORY., Ningbo, China | DC 12 V, Response time 20 ms |
| Nozzle | 40015 | Ningbo Licheng Agricultural Spray Technology Co., Ltd., Ningbo, China | Spray angle 40° |

## 2.2. Pressure Stabilization Design of the Herbicide Supply Unit

Mechanized transplanted open-field cabbage has 3 rows per ridge, with a row spacing of 45 cm and a plant spacing of 30 cm, and its growth cycle is 45~65 days. Based on the planting mode and growth characteristics of cabbage, a stable pressure herbicide supply unit, as shown in Figure 2, was designed in this study. There are 5 fan-shaped nozzles in the herbicide delivery system, the distance between nozzles is 22.5 cm, and the spray angle is 40°. Among them, nozzles 1, 3, and 5 are aimed at 3 rows of cabbage, and nozzles 2 and 4 are aimed at one ridge of cabbage. At a spraying height of 30 cm, the spraying range of the 5 nozzles can cover the whole cabbage field. During operation, driven by the engine power output shaft, the plunger pump sucks the liquid herbicide through the filter and pumps it into the pipeline.

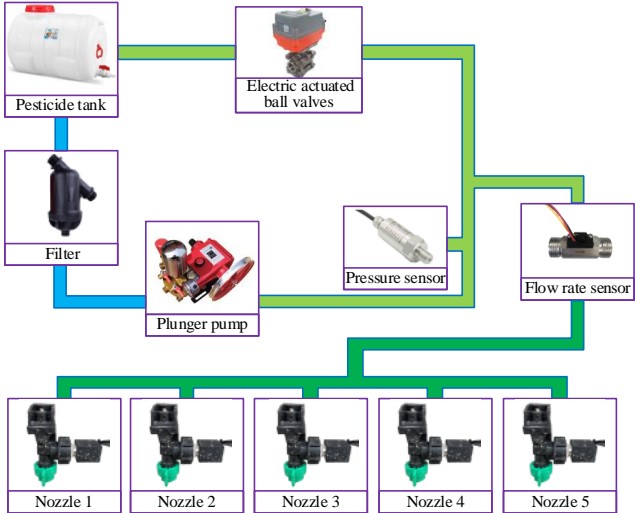

**Figure 2.** The overall configuration of the stabilized pressure spray system.

In this study, the driving system of the spraying machine and herbicide supply unit are the same engine. When the speed of the engine power output shaft changes, the pressure of the herbicide supply unit will fluctuate, and the spray amount will be affected. An

incremental PID control method is used to stabilize the pressure of the herbicide delivery system. The control voltage signal of the electric ball valve is DC 0~110 V, and the opening degree of the ball valve (0~100%) is proportional to the control voltage signal. The C37 controller adjusts the valve opening of the electric ball valve according to the difference between the spraying pressure obtained from the pressure sensor and the set spraying pressure to realize the stability of the spraying pressure. The control process is shown in Figure 3.

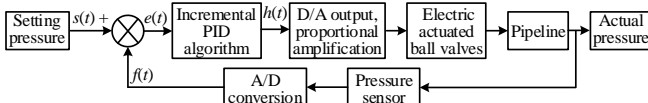

**Figure 3.** Spray pressure PID control process. Note: $s(t)$ is the set pressure, $f(t)$ is the feedback pressure, $e(t)$ is the difference between the system pressure and set pressure, $h(t)$ is the voltage expectation value for PID output, A/D is the conversion of analog to digital, D/A is the conversion of digital to analog.

The herbicide supply system was built according to the abovementioned PID control principle. The PID adjustment parameters were tuned to obtain the optimum values, and the proportional coefficient $k_p$, integral coefficient $k_i$, and differential coefficient $k_d$ were set to 0.12, 0.09, and 0.02, respectively. The mean stabilization time of the herbicide supply system with stable pressure was 3.2 s, and the deviation of pressure control was less than 0.05 MPa. During the spray process, the frequent opening and closing of solenoid valves (No. 1, No. 3, and No. 5 solenoid valves only open at the position between cabbage plants) will lead to periodic changes in the pressure of the herbicide supply unit. The results of previous experiments show that periodic changes cause small pressure fluctuations. In this study, the PID control object is the main pipeline pressure, and the control purpose is to stabilize the pressure of the system within a certain period of time.

### 2.3. System Software Design

The workflow of the intermittent weed spraying control system for open-field cabbage is shown in Figure 4. During operation, the C37 controller collects the pressure information of the herbicide supply unit in real time through a pressure sensor and controls the valve opening of the electric ball valve to adjust the system pressure according to the change in system pressure to realize herbicide supply unit pressure stability. Additionally, the C37 controller calculates the moving distance $s$ and working speed $v$ of the sprayer according to the pulse frequency and pulse number accumulated by the encoder. When the working speed exceeds 0.1 m/s, solenoid valves 1~5 are opened, and the sprayer starts spraying. The network camera installed at the front of the body of the spraying machine collects images in real time and sends the collected image information to the Jetson Xavier NX through a serial port. The upper computer software running on the Jetson Xavier NX performs target identification and location tracking and transmits the identified cabbage longitudinal canopy size $E$, which is the size of the cabbage canopy in the crop row direction, to the C37 controller through the CAN bus. The C37 controller stores this information and the cabbage position corresponding to the time stamp in the form of a message queue.

Once the nozzle reaches the designated cabbage position, the cabbage information at the corresponding position is called out, and the C37 controller sets the closing time of the solenoid valve according to the longitudinal canopy size $E$ to realize intermittent spraying. The real-time operation speed, spraying pressure, spraying flow, and accumulated spraying amount information are transmitted to the upper machine software through the C37 controller to realize real-time operation data display.

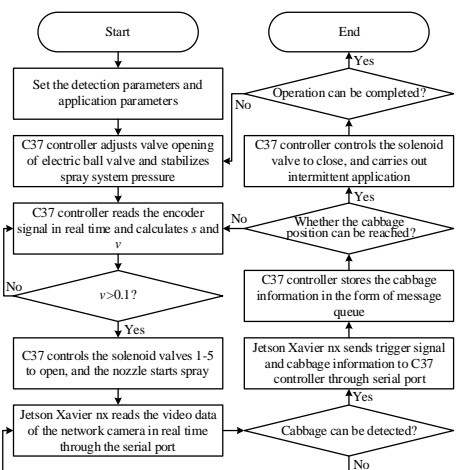

**Figure 4.** Workflow of intermittent weed spraying control system for open-field cabbage.

2.3.1. Communication Protocol

According to the requirements of independent control for opening and closing the 3 solenoid valves and the sensor data acquisition and standardizing the open-field cabbage spraying control communication, a communication protocol was formulated with reference to the ISO11783 protocol. Communication between the C37 controller and the upper computer software is carried out through the packet mode of the USB-CAN converter. The C37 controller receives the longitudinal canopy size information in real time through the USB-CAN converter, reads the sensor signal, and sends the sensor information to the upper computer software. The CAN message includes a frame header (1 byte), extended frame (1 byte), remote frame (1 byte), effective data length (1 byte), frame ID (4 bytes), and frame data (8 bytes). The communication protocol is shown in Table 2.

**Table 2.** Data field protocol of the herbicide bus system.

| Node | PDU Identification | Effective Data Length/Bytes | Frame Data Meaning |
|---|---|---|---|
| C37 | 0CE70001 | 2 | Data 0~1: Longitudinal canopy size of cabbage in row 1 |
| | 0CE70002 | 2 | Data 0~1: Longitudinal canopy size of cabbage in row 2 |
| | 0CE70003 | 2 | Data 0~1: Longitudinal canopy size of cabbage in row 3 |
| Jetson Xavier NX | 18E98384 | 6 | Data 0~1: Operation speed<br>Data 2~3: Spraying pressure<br>Data 4~5: Spraying flow rate |

The length of the longitudinal canopy size dataset in Table 1 is 2 bytes (unit: cm); the working speed data length is 2 bytes (unit: 0.01 m/s); the spraying pressure data length is 2 bytes (unit: 0.01 MPa); the spraying flow data length is 2 bytes in 0.01 L/min. The C37 controller only takes up 2 bytes for each received data frame and appends 00 when the valid data are less than 8 bytes. After the C37 controller program is started, the PDU identification of the message is evaluated. After the PDU identification is consistent, the Jetson Xavier NX data are read, the data are analyzed according to the communication protocol of the bus system to obtain the longitudinal canopy size, and the cabbage position is obtained according to the moving distance of the spraying machine at this time. The C37 controller controls the opening and closing of the solenoid valves according to the real-time operation speed, moving distance, and acquired cabbage information to realize intermittent weed spraying operation of open-field cabbage.

### 2.3.2. Control System Interface Design

In this study, real-time recognition and positioning of cabbage fields are carried out by video streaming. A cabbage-positioning method based on the Yolov5 model implanted with a transformer module proposed by researchers in a previous study is adopted in this paper [29], and a recognition model is developed. This recognition model has high recognition accuracy for cabbage under the condition of motion blur, and the recognition accuracy rate of cabbage is 96.1%. The recognition model is deployed on the Jeston Xavier NX development board, the image resolution is 480 pixels $\times$ 288 pixels, and the average image processing time is 51.07 ms.

Under a Linux system, the Python language is used to develop the upper machine interface of the open-field cabbage intermittent weed spraying control system, as shown in Figure 5, which is divided into an image display area, serial port control area, parameter display area, and equipment control area. The image display area can dynamically display the original image or processing results according to the need. The area is divided into three parts, each of which corresponds to nozzles 1, 3, and 5. The image display area includes a positioning line, and the recognition model outputs the boundary box coordinates of the target crop in the n-th frame image. The boundary box of the same target crop in the n-th frame is predicted by a Kalman filter algorithm, and the crop targets in the n-th and (n + 1)-th frames are associated with the Hungarian algorithm to realize the tracking of the target crop. When the cabbage crosses the set positioning line, the upper machine sends a trigger signal and sends the cabbage longitudinal canopy size *E* in the traveling direction to the C37 controller. The serial port control area is used for communication control between the upper machine and the lower machine. The parameter display area displays the speed, flow, and pressure information transmitted by the lower machine in real time. The equipment control area is used for network camera opening and closing and operation starting and stopping control.

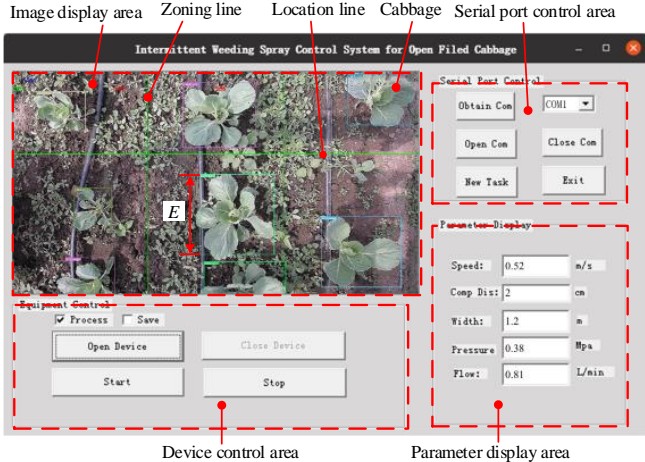

**Figure 5.** The operation interface of the intermittent weed-spraying control system in a cabbage field. Note: *E* in the figure refers to the longitudinal canopy size of the cabbage.

Before an operation, according to the growth height and size of the cabbage in the open field, the height of the network camera is adjusted and calibrated; then, the open-field cabbage intermittent weed spraying control system software is started, and parameters such as spraying pressure and spraying width are set, and the serial port is opened through the communication control area to realize communication between the upper computer and the lower computer. "Start job" is clicked to enter the real-time identification and cabbage processing state. The system obtains the current working speed of the spraying machine, opens solenoid valves 1~5 when the working speed exceeds 0.1 m/s, and evaluates the set cabbage identification and positioning conditions in real time. When the conditions are met, the upper computer sends the corresponding cabbage information to the C37 controller through the serial port, and the C37 controller controls the starting and stopping of the

spraying action according to the information and finally completes the weed spraying operation. Compared with a traditional continuous application method, the system can accurately identify the position and longitudinal canopy size of a cabbage, control the start and stop of spraying action through a solenoid valve, and spray only between cabbage rows and plants to reduce the amount of herbicide deposited on the cabbage. This improves the herbicide utilization rate and reduces herbicide residue in the cabbage.

## 3. Intermittent Weed Spraying Open-Field Cabbage Weed Spraying Control Method

There is a certain installation distance between the network camera and the nozzles. When the network camera detects cabbage, the information needs to be saved to the controller through the serial port, and the corresponding action can be made only when the sprayer moves to the designated position. According to the system schematic diagram in Figure 6, the delay compensation distance $l$ of the control system is determined by the system delay time $t$, the distance $l$ between the nozzle and the field of view positioning line of the upper machine, and the working speed $v$ of the weed sprayer. Among them, the system delay time comes from the cabbage identification time, communication time, solenoid valve response time and droplet settling time, and each part of the time can be obtained according to the experimental method in [30]. The calculation formula of the delay compensation distance of the control system is as follows

$$L = l - vt \tag{1}$$

where $L$ is the delay compensation distance of the control system, m; $l$ is the distance between the nozzle and the field of view positioning line of the upper machine, m; $v$ is the working speed of the sprayer, m/s; and $t$ is the system delay time, s.

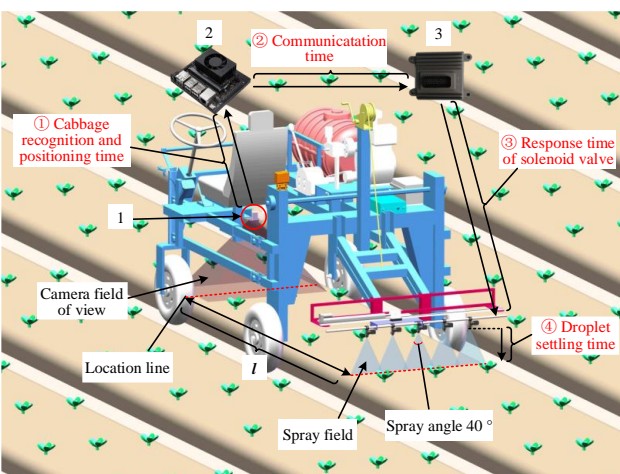

**Figure 6.** Schematic diagram of the field cabbage weed spraying system. Note: Serial number 1 in the figure is the network camera; serial number 2 is the Jetson Xavier NX; serial number 3 is the C37 controller.

To determine whether the nozzle reaches the target position accurately, the C37 controller needs to read the pulse signal of the encoder in real time. By calculating the product of the pulse frequency $f$ and the driving distance $q$ of the spraying machine corresponding to one pulse, the working speed $v$ of the spraying machine is obtained, and the moving distance $s$ of the weed spraying machine is calculated according to the number of pulse signals. When the number of encoder pulses $r$ reaches 10,000, the number of encoder pulses is reset to zero, counting is started again, and the counting value $c$ is increased by 1. The calculation formula is as follows:

$$v = fq \tag{2}$$

$$s = q(10000c + r) \tag{3}$$

where *s* is the moving distance of the spraying machine, m; *q* is the moving distance of the spraying machine corresponding to one pulse, m; *r* is the number of pulses; and *c* is the count value.

For the process of intermittent weed spraying of open-field cabbage, the C37 controller reads the pulse signal of the encoder in real time and calculates the moving distance *s* and working speed *v* of the spraying machine using Formulas (2)~(3). When a cabbage is detected, the Jetson Xavier NX sends a trigger signal through the serial port and sends the cabbage longitudinal canopy size *E* to the C37 controller. The C37 controller obtains the cabbage position by calculating the arithmetic sum of the moving distance *s* of the spraying machine and the delay compensation distance *L* at this time and stores the cabbage longitudinal canopy size *E* and the cabbage position in a message queue together. When the moving distance of the sprayer is larger than the cabbage position of the message queue, it indicates that the intermittent weed sprayer has reached the target cabbage position, the C37 controller controls the solenoid valve to close, and the closing time is determined according to the longitudinal canopy size *E* of the target cabbage, thus realizing the intermittent cabbage weed spraying operation. After the current position, breakpoint spraying is completed, the current detection position information is released, and the solenoid valve is reopened. When the closing condition of the solenoid valve is met, the intermittent cabbage weed spraying operation is completed according to the above process. In addition, through the pre-experiment, there are spraying errors in the operation of the system, and the average spraying error is 3 cm, which makes the liquid herbicide cover of the inter-plant position of cabbage plants incomplete, and the effective spraying coverage rate low. The effective spraying coverage rate affects the actual weeding operation quality of the system. To reduce the influence of spraying error on the effective spraying coverage, the longitudinal canopy size compensation was set so that the closing time of the solenoid valve was set according to the target longitudinal canopy size *E*, thus shortening the closing time of the solenoid valve at the cabbage position and improving the effective spraying coverage. In this study, the compensation of the longitudinal canopy size is called the offset, and the value is 0~3 cm, verified by experiments. The longitudinal canopy size compensation is shown in Figure 7.

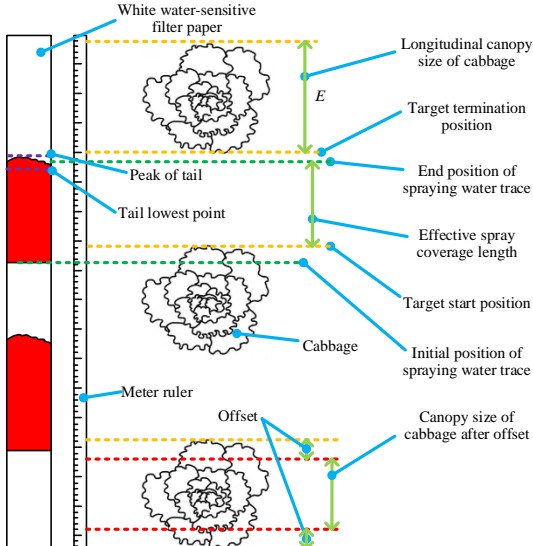

**Figure 7.** Schematic diagram of compensation for the longitudinal canopy size and weed spraying operation of cabbage. Note: It was found in the experiment that the liquid herbicide may exhibit a tailing phenomenon when sprayed on white water-sensitive filter paper, so the average of the lowest and highest points at the tail of the spraying water trail was used as the boundary position of the spraying water trail. The starting and ending positions of the target in the picture are the edge positions of the cabbage canopy.

## 4. Experiment and Results Analysis

### 4.1. Offset Selection for Cabbage Weed Spraying

To determine the offset for cabbage weed spraying, an experiment was carried out on 22 July 2022 (cloudy, wind direction 249~316°, wind speed 0.67~1.16 m/s, daytime temperature 27~31 °C) at the National Precision Agriculture Research Demonstration Base in Xiaotangshan Town, Changping District, Beijing. With reference to the agronomic open-field cabbage planting, two rows of potted cabbage were set up in the experiment, with a row spacing of 45 cm, plant spacing of 30 cm, and 25 cabbage plants in each row. White water-sensitive filter paper (model CFY-05, Shenzhen Chuangbaoda Technology Co., Ltd., Shenzhen, China) was arranged beside each row of cabbages. This filter paper quickly turned red when exposed to water, and nozzles 2~3 on the movable spray rod were aligned with each piece of water-sensitive filter paper. The experiment was carried out at a low sprayer I gear speed. During the experiment, the valve opening of the electric ball valve was adjusted, and the initial working water pressure was set to 0.4 MPa. The spraying height was adjusted to 30 cm using the spraying machine hand winch, and three parallel experiments were carried out along the crop row under the condition of an offset of 0~3 cm. Each parallel experiment was repeated three times. A diagram of the intermittent weed spraying operation is shown in Figure 7. After each experiment, photographs of each experiment result were taken, and experimental data were obtained by reading the measurement indexes on the meter scale, shown in Table 3.

**Table 3.** Measurement indicators.

| Variable | Abbreviation | Interpretation | Unit |
|---|---|---|---|
| Spraying error | SE | The distance between the center of filter paper spraying water trace and the target center in the traveling direction. | cm |
| Average absolute error | MAE | Average of absolute values of all spraying errors in a set of data. | cm |
| Root mean square error | RMSE | In a set of data, the square root of the mean value of the sum of squares of all errors to the target. | cm |
| Effective spraying cover length | ESCL | Traveling direction filter paper sprays water traces to cover the part of the target length. | cm |
| Effective spraying coverage rate | ESCR | Ratio of effective spraying cover length to target length. | % |
| Average effective spraying coverage rate | AESCR | Average effective spraying coverage rate. | % |
| Spraying coverage cabbage rate | SCCR | Ratio of cabbage length covered by spraying water trace to total cabbage length. | % |
| Average spray cabbage coverage rate | ASCCR | Average cabbage spraying coverage. | % |

The experimental sites with different offsets are shown in Figure 8. In the 0~1 cm offset experiment, due to the spraying error in the system, a spray leak phenomenon appeared. In addition, in the 3 cm offset experiment, some cabbage positions were sprayed by mistake. By analysis, it is argued that due to the limitation of the response time of spraying action in the system, under the condition of a 3 cm offset and the same cabbage canopy size, the solenoid valve may not be closed due to the short closing time at the cabbage position, resulting in false spraying.

**Table 4.** Spray results with different offsets.

| Offset/cm | MAE/cm | RMSE/cm | AESCR/% | ASCCR/% |
|:---:|:---:|:---:|:---:|:---:|
| 0 | 2.88 | 3.80 | 80.1 | 15.9 |
| 1 | 3.17 | 3.78 | 88.8 | 22.6 |
| 2 | 2.87 | 3.40 | 98.4 | 28.3 |
| 3 | 3.12 | 3.78 | 100 | 50.8 |

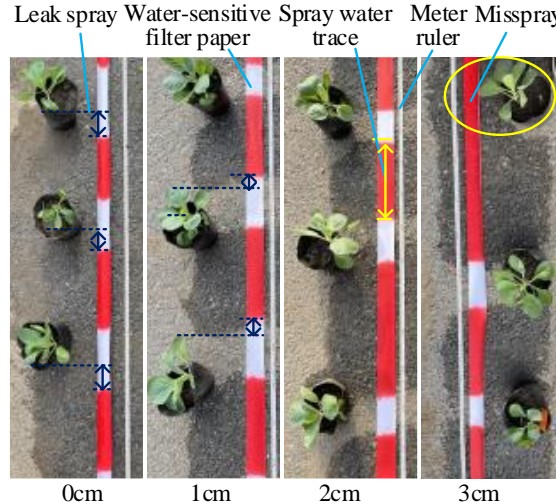

**Figure 8.** Experimental results with different offsets. The spraying experiment results with different offsets are shown in Table 4. According to statistics, in the experiment with an offset of 0~3 cm, the average actual vehicle speed is 0.51 m/s.

In all the offset experiments, the MAE and RMSE have no obvious changes. In the spraying process without distance compensation (the offset is 0 cm), the AESCR is only 80.1%, which will lead to the spraying being unable to cover the area between cabbage plants completely and affect the actual weeding operation quality. In Figure 8, the leakage caused by setting the offset to 0 cm also reflects this phenomenon. With the increase in offset, the AESCR and ASCCR increase. When the offset exceeds 2 cm, the AESCR is close to the critical value of 1. Through analysis, it is argued that the SE is basically unchanged under the condition of constant velocity, and the spraying length increases with increasing offset, which leads to an increase in the AESCR and ASCCR. In addition, when the offset is 3 cm, due to the limitation of the response time of the spraying action, the solenoid valve may not be closed at the cabbage position, resulting in accidental spraying. Therefore, on the premise of satisfying the weed spraying operation, a compensation length of 2 cm was selected; consequently, the AESCR was 96.8%, and the ASCCR was 28.3%.

*4.2. Effect of Speed on the Accuracy of Intermittent Cabbage Weed Spraying*

To verify the influence of speed on the accuracy of the intermittent weed spraying system, an experiment was carried out on 23 July 2022 (sunny, wind direction 290~338°, wind speed 0.52~0.95 m/s, daytime temperature 25~32 °C) at the National Precision Agriculture Research Demonstration Base in Xiaotangshan Town, Changping District, Beijing. During the experiment, by adjusting the valve opening of the electric ball valve, the initial working water pressure is set to 0.4 MPa. The spraying height was adjusted to 30 cm using the spraying machine hand winch, and three parallel experiments were carried out along the crop row at three different speeds of the weed spraying machine, namely, I-gear low speed, I-gear high speed, and II-gear low speed. The site of the experiments for weed spraying accuracy at different speeds is shown in Figure 9a.

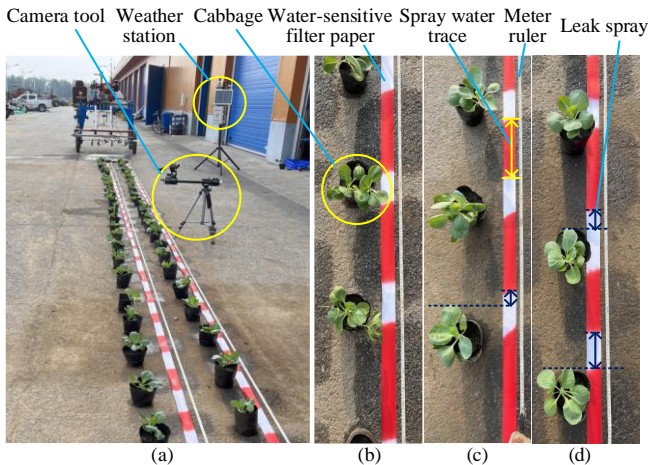

**Figure 9.** Accuracy experiment for weed spraying at different speeds. (**a**) Experiment site; (**b**) 0.51m/s; (**c**) 0.68m/s; (**d**) 0.80m/s.

The experimental results are shown in Table 5. The MAE and RMSE increased with increasing operation speed, which showed that spraying accuracy decreased with increasing operation speed. When the spraying accuracy decreases, the probability of liquid herbicide deposition in the cabbage will increase; that is, the ASCCR will increase. Under the same target spraying size, the probability of liquid herbicide deposition in the cabbage canopy increased while the corresponding AESCR decreased, which showed that the ASCCR and AESCR were negatively correlated. The upwards trend of spraying leakage at speeds of 0.51, 0.68, and 0.80 m/s in Figure 9 also reflected this phenomenon. In addition, when the working speed was 0.51 m/s, the MAE and RMSE were the smallest, which were 2.87 cm and 3.40 cm, respectively, and the AESCR was 98.4%. When the driving speed was 0.77 m/s, the MAE and RMSE were the largest, while the AESCR was the lowest. The experimental results of the spray accuracy for weeding at different speeds showed that the MAE, RMSE, and ASCCR increased, the AESCR decreased, and the accuracy of weeding and spraying decreased with increasing working speed. At a working speed of 0.51 m/s, the MAE and RMSE of the weed spraying were less than 2.87 cm and 3.40 cm, respectively, and the AESCR was 98.4%, which verified the feasibility of the intermittent cabbage weed spraying operation.

**Table 5.** Experimental results of intermittent weed spraying.

| Operation Speed/(m·s$^{-1}$) | MAE/cm | RMSE/cm | AESCR/% | ASCCR/% |
|---|---|---|---|---|
| 0.51 | 2.87 | 3.40 | 98.4 | 28.3 |
| 0.68 | 5.20 | 6.19 | 79.2 | 36.6 |
| 0.80 | 8.55 | 9.67 | 56.6 | 67.5 |

### *4.3. Field Experiment*

To determine the actual operation quality of the intermittent weed spraying system for open-field cabbage, an experiment was carried out on 17 September 2022 (cloudy, wind direction 173~216°, wind speed 1.03~1.60 m/s, daytime temperature 25~30 °C) in a cabbage field of the Xiaotangshan National Precision Agriculture Research Demonstration Base, Changping District, Beijing. Cabbage was planted in 3 rows per ridge; the cabbage row spacing was 45 cm, the plant spacing was 30 cm, and the total row length was 60 m.

#### 4.3.1. Intermittent Cabbage Weed Spraying in Open Field Accuracy Experiments

Before the experiment, the spraying height was adjusted to 30 cm, and white water-sensitive filter paper was arranged beside the first row of cabbage only (the No. 1 nozzle on the movable spray rod was aligned with the white water-sensitive filter paper). During

the experiment, by adjusting the valve opening of the electric ball valve, the initial working water pressure was set to 0.4 MPa. In addition, the experiment was only carried out at the low I gear speed of the sprayer, the experiment was repeated three times, and the length of each experiment was 20 m. The accuracy experiment for weed spraying of open-field cabbage is shown in Figure 10, and the measurement of each experiment index is consistent with the above laboratory experiment.

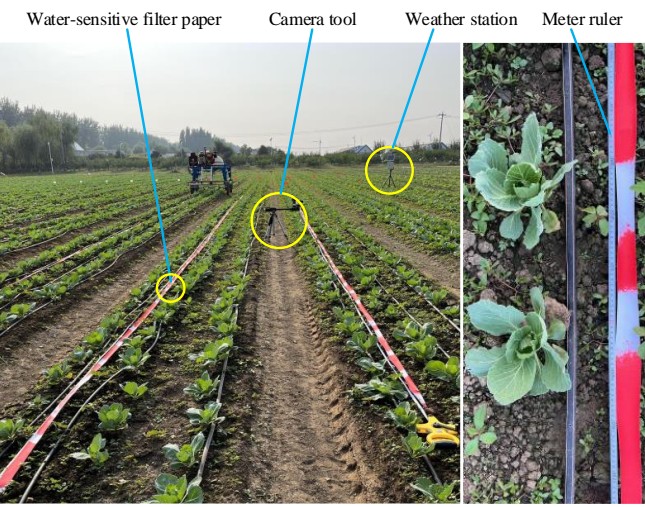

**Figure 10.** Accuracy experiment for weed spraying for the open-field cabbages.

According to the experimental statistics, the actual average speed during the operation was 0.54 m/s. According to the measurement method of laboratory experiment-related indexes, the MAE, RMSE, AESCR, and ASCCR of the three intermittent weed sprayings were calculated to be 3.53 cm, 3.98 cm, 96.2%, and 32.5%, respectively.

The experimental results on the accuracy of intermittent weeding and spraying for open-field cabbage showed that the AESCR reached 96.2% when the operation speed was less than 0.54 m/s, which meets the requirements of intermittent weed spraying of cabbage.

4.3.2. Comparative Experiment of Efficacy between Intermittent Weed Spraying and Constant-Rate Application of Cabbage in an Open Field

In this study, herbicide residues and the weed-killing rate were used as the indexes of an efficacy comparison experiment between intermittent weed spraying and constant-rate application to cabbage in an open field. Twelve ridges of cabbage were set up in the experiment, of which six ridges were used for intermittent weed spraying and six for constant-rate application, and the experiment was carried out with an I-gear low-speed spraying machine. The herbicide was a combination used for weeding after cabbage seedlings. The effective ingredients were high-efficiency haloxyfop-methyl (108 g/L, in EC form, Jinan Kehai Co., Ltd., Jinan, China) and amchloropyridine (the total effective ingredient content is 30%, of which clopyralid and picloram account for 24% and 6%, respectively, in an aqueous solution, Hunan Bide Biochemical Technology Co., Ltd., Yueyang, China), Each set of herbicides mixed with 15 kg water was used to control annual gramineous weeds and annual broad-leaved weeds and was applied at the 3~5 leaf stage of weeds. After the application was completed, the amount of the two application methods was measured by weighing, and the herbicide-saving rate was calculated.

According to the weighing statistics before and after spraying the two weeding methods, as shown in Figure 11, the application amount of the continuous weeding method was 37.5 L, and the application amount of intermittent weed spraying was 26.9 L. Compared with the constant-rate application, the herbicide-saving rate of intermittent weed spraying was 28.3%.

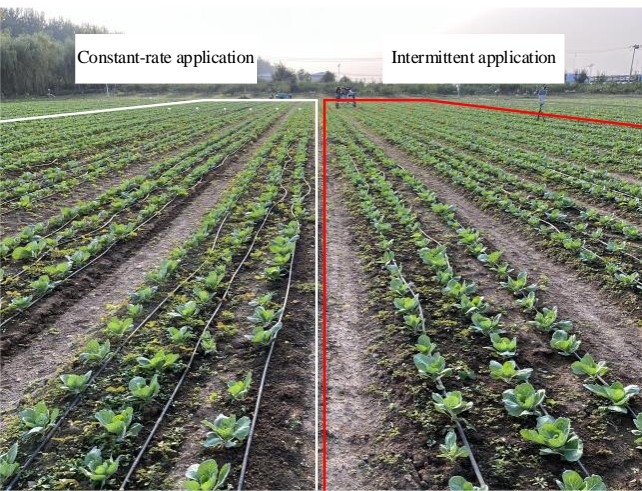

**Figure 11.** A comparative experiment of intermittent application and constant-rate application.

For field sampling of herbicide residues by intermittent weed spraying and constant-rate application, refer to NY/T 788-2018 and Standard Operating Procedures for Field Trial of Pesticide Registration Residues [31,32]. After a 2 h application, 12 normal and disease-free cabbage individuals and 12 surface soil samples (soil samples were collected between rows and plants of cabbage, 1 kg of soil samples were collected at each sampling point) were randomly collected in intermittent weeding and spray experiment areas and constant-rate application experiment areas, respectively, but no samples were collected within 0.5 m of the field or edge. After the collection was completed, it was transported back to the laboratory within 2 h and prepared into laboratory samples for cryopreservation. The sampling diagram of cabbage and soil is shown in Figure 12. Referring to GB/T 20769-2008 and GB 23200.109-2018 [33,34], herbicide residues in soil and cabbage were determined by liquid chromatography-tandem mass spectrometry (LC–MS/MS). Using the herbicide residue concentration (μg/kg) as the response variable and the weeding method as the independent variable, a one-way analysis of variance (ANOVA) was used to evaluate the difference in herbicide residue concentration in cabbage and soil. The hypotheses of ANOVA, such as independent observation value, normal distribution, and intragroup homovariance, were evaluated. Finally, the mean value and standard deviation of the herbicide residue concentration of the two weeding methods were calculated.

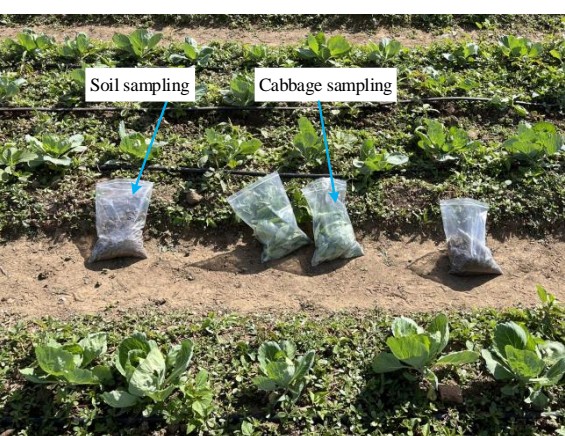

**Figure 12.** Schematic diagram of cabbage and soil sampling.

The results of the one-way ANOVA of herbicide residues in cabbage and soil by two weeding methods are shown in Table 6. The concentrations of clopyralid and picloram in cabbage were significantly different (*p* value < 0.01) for the two different application methods, which indicated that the different application methods indeed caused differences

in herbicide residues in cabbage. Similarly, there was no significant difference in the concentrations of clopyralid and picloram in soil for the two different application methods, and the $p$ values were 0.81 and 0.87, respectively (there was no significant difference when the $p$ value > 0.05), which indicated that there was no significant difference in herbicide efficacy at the location where weeds appeared under the two different application methods.

**Table 6.** Results of one-way analysis of variance (ANOVA) to compare the effects of two different weeding methods (intermittent application and constant-rate application) on herbicide residues in cabbage and soil.

| Position | Type of Pesticide | Source | Degree of Freedom DF | Mean Squares | F | *p* Value |
|---|---|---|---|---|---|---|
| Cabbage | 3,6-Dichloropicolinic acid | Intergroup | 1 | 42,771,864.8 | 45.27 | <0.01 |
| | | Intragroup | 22 | 944,890.7 | | |
| | | Total | 23 | | | |
| | Aminopyralid | Intergroup | 1 | 1,388,370.4 | 38.73 | <0.01 |
| | | Intragroup | 22 | 35,852.2 | | |
| | | Total | 23 | | | |
| Soil | 3,6-Dichloropicolinic acid | Intergroup | 1 | 15,780.9 | 0.06 | 0.81 |
| | | Intragroup | 22 | 267,335.6 | | |
| | | Total | 23 | | | |
| | Aminopyralid | Intergroup | 1 | 235.1 | 0.03 | 0.87 |
| | | Intragroup | 22 | 7957.0 | | |
| | | Total | 23 | | | |

The herbicide residue concentrations of the two weeding methods in cabbage and soil are shown in Table 7. Compared with the constant-rate application, the residual concentrations of clopyralid and picloram in cabbage by the intermittent application were reduced by 67.3% and 65.8%, respectively, and those in soil were reduced by 5.3% and 3.4%, respectively. The experimental results showed that the intermittent weed spraying system meets the requirements of weed spraying in open fields, and compared with the constant-rate application method, the herbicide residue concentration in cabbage decreased by 66.6% on average, and there was no significant difference in herbicide residue concentration in soil.

Under the intermittent and constant-rate application methods, there was little difference in the detection results of the proportion of the two herbicide components in different positions (cabbage, soil) and different application methods. The average residual concentration of picloram was approximately 81% lower than that of clopyralid, which is similar to the factory data of herbicide manufacturers (clopyralid accounts for 24% and picloram accounts for 6% of the effective components in the stock solution of picloram) and verifies the accuracy of the herbicide residue concentration detection experiment.

Fourteen days after the completion of the operation, the weeds were counted [35]. Referring to GB/T 17980.127-2004 [36], six plots were randomly selected from the two application methods; the plot size was 1.2 m (ridge width) × 1 m, and the weed coverage of each plot before and 14 days after the experiment was evaluated by photographs., Photoshop software was used to label cabbage in each image and separate cabbage from the image, to avoid the influence of cabbage on statistical weed coverage. Then, a super green algorithm was used to process each image after removing the cabbage, and the weeds were separated from the background. Finally, the proportion of weed pixels in each image was counted to obtain weed coverage, and the weed-killing rate was obtained by calculating the difference in weed coverage before and after application. The weed density statistics are shown in Figure 13.

**Table 7.** Statistics of herbicide residue concentration (μg/kg) in cabbage and soil using two weeding methods.

| Position | Type of Pesticide | Experiment | Mean ± SD | Minimum | Maximum | % Saving |
|---|---|---|---|---|---|---|
| Cabbage | 3,6-Dichloropicolinic acid | Intermittent application | 1294.5 ± 168.4 | 547.5 | 2979.4 | 67.3 |
| | | Constant-rate application | 3964.5 ± 340.6 | 2341.1 | 6737.9 | NA |
| | Aminopyralid | Intermittent application | 249.9 ± 33.0 | 159.0 | 595.1 | 65.8 |
| | | Constant-rate application | 730.9 ± 66.3 | 451.2 | 1214.6 | NA |
| Soil | 3,6-Dichloropicolinic acid | Intermittent application | 924.3 ± 117.0 | 244.4 | 1512.1 | 5.3 |
| | | Constant-rate application | 975.6 ± 165.5 | 291.4 | 2738.5 | NA |
| | Aminopyralid | Intermittent application | 181.3 ± 25.2 | 51.6 | 326.2 | 3.4 |
| | | Constant-rate application | 187.6 ± 24.0 | 43.9 | 410.0 | NA |

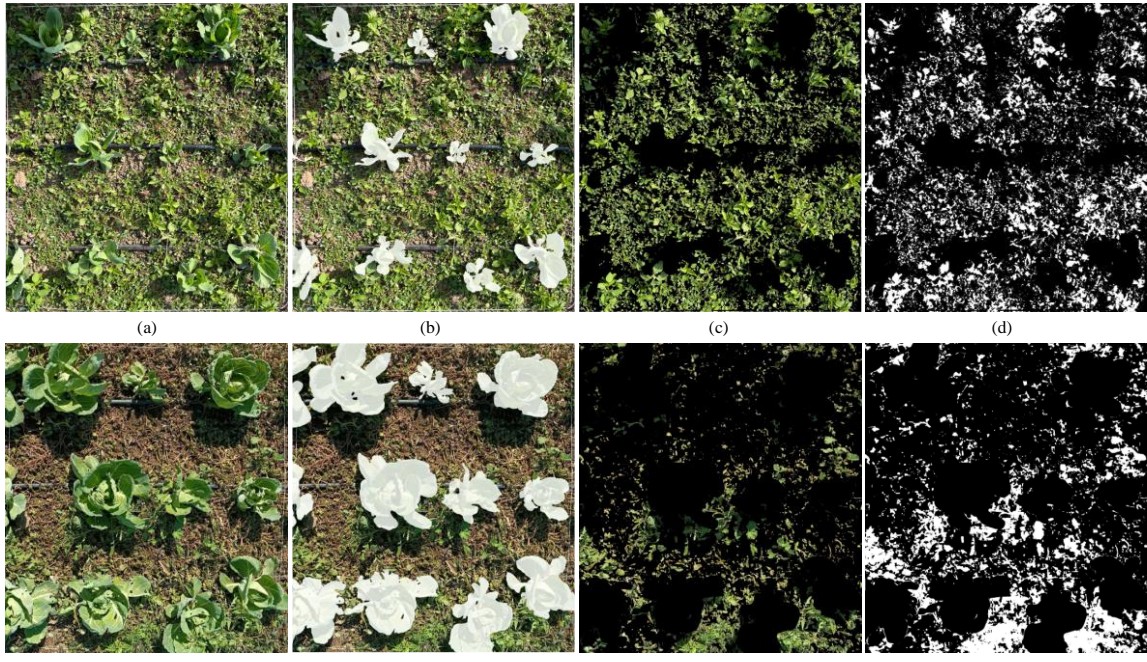

**Figure 13.** Statistical indication of weed coverage. (**a**) Original image before pesticide application; (**b**) Remove cabbage image before pesticide application; (**c**) Super green algorithm processing image before pesticide application; (**d**) Binary weed image before pesticide application; (**e**) Original image after pesticide application; (**f**) Remove cabbage image after pesticide application; (**g**) Super green algorithm processing image after pesticide application; (**h**) Binary weed image after pesticide application.

The weed-killing results of the two weeding methods are shown in Table 8. The average weed-killing rates of intermittent and continuous herbicide spraying were 94.8% and 96.8%, respectively, and the average weed coverage was 2.1% and 1.3% 14 days after herbicide spraying. The results showed that the system had a good weeding effect, which was close to the continuous herbicide spraying method and met the needs of intermittent herbicide spraying operation of open-field cabbage. Under the two weeding methods, the results of similar weed-killing rates and the results of no significant difference in herbicide residue concentration in soil are mutually confirmed.

**Table 8.** Weed-killing results of the two weeding methods.

| No./Weed Coverage | Intermittent Application | | | Constant-Rate Application | | |
|---|---|---|---|---|---|---|
| | Before Application | After Application | Weed Killing Rate | Before Application | After Application | Weed Killing Rate |
| 1 | 39.8 | 2.1 | 94.7 | 44.0 | 1.4 | 96.9 |
| 2 | 38.6 | 2.2 | 94.4 | 45.2 | 1.6 | 96.5 |
| 3 | 36.5 | 1.6 | 95.8 | 35.9 | 0.9 | 97.5 |
| 4 | 38.4 | 2.0 | 94.9 | 39.9 | 1.7 | 95.7 |
| 5 | 43.9 | 2.3 | 94.9 | 36.7 | 1.1 | 97.1 |
| 6 | 43.3 | 2.6 | 94.0 | 45.9 | 1.3 | 97.2 |
| Mean value | 40.1 | 2.1 | 94.8 | 41.3 | 1.3 | 96.8 |

**5. Discussion**

At present, relevant studies either look at two indexes, namely, herbicide saving rate and deposition density [25], or one index, system operation accuracy [37], to verify the system performance without considering the actual spraying effect under operation accuracy. The important purpose of precise weed spraying is to achieve effective weed control. Based on the meeting of the requirements of herbicide application accuracy, it is of great research significance to analyze further the actual efficacy under precise weeding operation and track the amount applied and residue of precise weed spraying. In this paper, the spraying control system and spraying operation of intermittent weeding were evaluated for four aspects: spraying accuracy, herbicide saving rate, herbicide efficacy, and herbicide residue. In the face of the lack of relevant operation specifications and national standards, this paper provides a relatively perfect experiment and evaluation method for this operation mode.

The pesticide-saving rate of weeding operations is related to crop agronomy and canopy size [38]. For example, S. A. Shearer et al. [39] saved 15% in spot weeding of soybean crops. K-H DAMMER et al. [40] used fixed-point spraying technology in carrot crops to save 30–40% of herbicides. In this paper, under the agronomic planting conditions of cabbage canopy size ranging from 15 to 30 cm, row spacing 45 cm, and plant spacing 30 cm, compared with continuous spraying, intermittent spraying saved 28.3% of pesticides, which not only met the needs of the weeding operation demand but also made a great contribution to cost savings.

The precision of intermittent spraying operation is related to canopy size. Because there are many kinds of weeds, the color discrimination between weeds and crops is low, and it is difficult to identify them. Moreover, it is more difficult to identify weeds at the seedling stage. Compared with weeds, crops are a single species, and the shape characteristics tend to be consistent in the same period. From the perspective of identifying cabbage, this paper proposes an intermittent weed spraying control method for open-field cabbage, which integrates the cabbage position, the cabbage canopy size, and the operation speed of the spraying machine and effectively improves the recognition efficiency.

When the speed is less than 0.51 m/s, the average absolute error is 2.87 cm, which is 24.5% higher than that of 3.80 cm at the same speed in reference [35]. However, during the experiment, it was found that when the working speed increases, the vibration amplitude of the car body increases due to the increase in the rotating speed of the single-cylinder engine and the uneven ground, which leads to an increase in the spraying error and affects the actual spraying effect. An electric driving walking system can be adopted and adapted to integrate satellite differential positioning navigation, which can improve the stability of the walking speed and improve the positioning accuracy using absolute coordinates.

The deposition of herbicides on crops causes herbicide residue, which poses a great threat to the growth and development of crops. At the same time, herbicide residue in crops means the safety of agricultural products is not guaranteed. The weed-killing rate of constant-rate application was 96.8%, and the average weed-killing rate of intermittent

application was 94.8%. Compared with the constant-rate application method, the herbicide efficacy of the intermittent weed spraying method was similar, the herbicide residue in cabbage was greatly reduced, and the herbicide residue concentration in cabbage was reduced by 66.6% on average. Thus, intermittent weed spraying has important research significance and application value to ensure the safety of agricultural products. In theory, herbicide residue in cabbage can be reduced by nearly 100%, but in fact, there are herbicide drift and spraying errors, which means that it cannot reach 100% in practice. We can further reduce crop herbicide residue through drift prevention and control, improve spraying accuracy and make more contributions to ensuring the normal growth of crops and the safety of agricultural products.

## 6. Conclusions

(1)  The field environment is complex, and weeding by identifying weeds faces problems such as a wide variety of weeds and overlapping clusters, resulting in low identification accuracy. Compared with weeds, crops are a single species, and their appearance characteristics tend to be consistent in the same period. Therefore, an intermittent weed spraying control method for open-field cabbage is proposed that integrates the cabbage position and canopy size and the working speed of the spraying machine. A steady pressure spray system and an intermittent weed spraying control system are established. In addition, experimental verification was carried out through measurement indexes such as the spraying precision, herbicide saving rate, herbicide efficacy and herbicide residue. Since the industry is currently faced with a status quo of a lack of relevant operational norms and national standards for the precise weed spraying operation mode, this paper provides a relatively perfect experiment and evaluation method for this operation mode.

(2)  Operating speed is the main factor affecting the precision of intermittent weeding spraying. With increasing working speed, the MAE, RMSE, and ASCCR increase, the AESCR decreases, and the accuracy of the weed spraying decreases. When the working speed is 0.51 m/s, the MAE and RMSE are not higher than 2.87 cm and 3.40 cm, respectively, and the AESCR is 98.4%, which verifies the feasibility of the intermittent cabbage weed spraying operation.

(3)  The average weed-killing rate of intermittent weed spraying in open-field cabbage is 94.8%, and the herbicide saving rate can reach 28.3% under the condition of a similar weeding effect to that of constant-rate application, which not only meets the needs of intermittent weed spraying in open-field cabbage but also has great significance for improving the herbicide utilization rate. At the same time, it also made a great contribution to cost savings.

(4)  The pesticide residues in cabbage and soil under intermittent weed spraying and constant-rate application were compared. Compared with a constant-rate application, there was no significant difference using intermittent weed spraying in herbicide residue concentration in soil, and the herbicide residue concentration in cabbage detected decreased by 66.6% on average, which has important research significance and application value for ensuring the normal growth of crops and the safety of agricultural products.

**Author Contributions:** Conceptualization, S.Z. and X.Z. (Xueguan Zhao); Methodology, S.Z., C.Z., K.Y. and X.Z. (Xinwei Zhang); Validation, S.Z. and X.Z. (Xinwei Zhang); Formal analyses, S.Z. and X.Z. (Xinwei Zhang); Investigation, S.Z. and H.F.; Resources, C.Z., K.Y. and X.Z. (Xueguan Zhao); Data curation, S.Z.; Writing—original draft, S.Z.; Writing—review and editing, X.Z. (Xueguan Zhao) and C.Z.; Funding acquisition, C.Z. and X.Z. (Xueguan Zhao); Supervision, X.Z. (Xueguan Zhao), K.Y. and C.Z. All authors have read and agreed to the published version of the manuscript.

**Funding:** Support was provided by (1) the National Natural Science Foundation of China Youth Science Foundation Project (32201647); (2) the National Modern Agricultural Industrial Technology

System Project (CARS-23-D07); and (3) the Open Project of Intelligent Equipment Research Center, Beijing Academy of Agriculture and Forestry Sciences (KF2020W010, KFZN2021W001).

**Data Availability Statement:** The data presented in this study are available upon request from the corresponding author.

**Conflicts of Interest:** The authors declare no conflict of interest.

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
