# Peer review of "Design of an Intermittent Herbicide Spray System for Open-Field Cabbage and Plant Protection Effect Experiments"

_agronomy, doi:10.3390/agronomy13020286_

Round 1

Reviewer 1 Report

Point 1: The cabbage target identification method from the previous study can be briefly described in the introduction to add to the integrity of the thesis

Point 2: The height and size of cabbage varies from region to region, fertility period to fertility period and management style. It is tedious to adjust the height of the webcam before each weeding operation and it is recommended to determine a universal height

 Point 3: How are weeds under the canopy cover of cabbage considered and do they need to be applied? Is it possible to solve this problem by lowering the height of nozzles 2 and 4 between rows and increasing the spray angle?

 Point 4: Figure 6 is slightly cluttered with confusing arrows, please indicate the system components and response times in order and label them with serial numbers

 Point 5: In lines 58-59,the liquid sprayed on the plants will cause waste, which will affect the normal growth of crops and herbicide residues.the expression is improper.

 Point 6: In line 304, which makes the liquid herbicide not completely cover the cabbage plants,Why herbicides cover cabbages?

Reviewer 2 Report

I reviewed the manuscript title “Design of an Intermittent Herbicide Spray System for Open-Field Cabbage and Plant Protection Effect Experiments” submitted by Zheng et al for possible publication in Agronomy journal. The performed work comes within the scope of the journal.  The manuscript is well written and can be considered for possible publication after Major revision. The main issue with the paper is the writing style, the authors have written too big sentences, it is suggested that concise the sentences. Besides, there are many sentences in the manuscript which are repeated too many times, thus it is suggested paper should be proof read from the native English.

Why the area was divided into three parts? and on which factors the number of nozzles was considered?

Better to mention completed information about used items such as nozzle type and its discharge etc.

Authors said that the height of the camera was adjusted, so can you what was the height of the cabbage during experiment and you selected height. Was it same as spraying height?

Improve the figures quality and figure 7 and Table 3 can be deleted.

What do you mean by sprayer I gear speed?

Reviewer 3 Report

The article proposes an intermittent method for spraying cabbage in the open field. The article is interesting and relevant. To improve the quality of the article, the following changes should be made:

1. More useful conclusions should be drawn. Perhaps the authors should rewrite the conclusions more analytically.

2. Section 6. "Conclusion" should also clearly describe the novelty of the research.

3. It would be useful to provide economic data  and compare it with some similar systems to improve the quality of work.

4. How does the developed system account for distortions arising from droplet evaporation, gravitational separation of droplets by size (deposition rate), inertia of spray mechanisms, method and exposure time?

5. A section on the prospects for further research should be added, in which the ways of solving the identified shortcomings are described.

6. It is necessary to describe the method of plant identification in more detail, add the speed characteristics of image processing.

Round 2

Reviewer 2 Report

Manuscript can be accepted for publication.